# Prognostic Impact of YKL-40 Immunohistochemical Expression in Patients with Colorectal Cancer

**Il Hwan Oh [1,†], Jung-Soo Pyo [2,†] and Byoung Kwan Son [1,*]**

[1] Department of Internal Medicine, Uijeongbu Eulji Medical Center, Eulji University School of Medicine, Uijeongbu-si 11759, Korea; 20180121@eulji.ac.kr

[2] Department of Pathology, Uijeongbu Eulji Medical Center, Eulji University School of Medicine, Uijeongbu-si 11759, Korea; jspyo@eulji.ac.kr

[*] Correspondence: sbk1026@eulji.ac.kr; Tel.: +82-31-951-2130

[†] Il Hwan Oh and Jung-Soo Pyo have contributed equally to this study.

**Abstract:** This study aims to examine the clinicopathological and prognostic significance of the YKL-40 immunohistochemical expression of tumor and immune cells through human colorectal cancer (CRC) tissue. We performed immunohistochemistry for YKL-40 and investigated the clinico-pathological and prognostic impact of the YKL-40 expression of tumor (T-YKL-40) and immune cells (I-YKL-40) in CRC. We also evaluated the correlation between YKL-40 and PD-L1 expression and the immunoscore. YKL-40 was expressed in 22.6% and 64.2% of T-YKL-40 and I-YKL-40, respectively, out of the 265 CRC tissues. The I-YKL-40 expression significantly correlated with well and moderately differentiated tumors. The PD-L1 expression in immune cells significantly correlated with the I-YKL-40 expression, but not T-YKL-40 expression ($p = 0.020$ and $p = 0.846$, respectively). The I-YKL-40 expression significantly correlated with a worse overall survival rate but not recurrence-free survival ($p = 0.047$ and $p = 0.080$, respectively). However, there was no significant correlation between the T-YKL-40 expression and survival. In CRCs with a high immunoscore, patients with I-YKL-40 expression demonstrated worse overall and recurrence-free survival than those without I-YKL-40 expression. Our results demonstrated that I-YKL-40 expression significantly correlated with tumor differentiation and PD-L1 expression in immune cells. I-YKL-40 expression can be useful for the prognostic stratification of CRC patients.

**Keywords:** colorectal cancer; YKL-40; immunohistochemistry; immunoscore; prognosis





## 1. Introduction

Colorectal cancer (CRC) is still one of the most common tumors worldwide, although effective multimodality treatment has improved the survival rate [1]. Moreover, high heterogeneity in CRC is widely known to affect tumor progression and treatment resistance [2]. This tumor heterogeneity is driven by cell-to-cell interactions in the tumor microenvironment (TME), which consists of tumor cells, stromal tissue, and extracellular matrix components [3]. YKL-40 is a chitinase, like a glycoprotein, expressed by various cell types, such as macrophages, neutrophils, vascular smooth cells, epithelial cells, chondrocytes, and synoviocytes [4].

The correlation between YKL-40 and cancers and chronic inflammatory diseases has been reported [4]. YKL-40 is considered to play a role in cancer through the process of inflammation surrounding the tumor cells, angiogenesis, and extracellular matrix remodeling [5,6]. According to a recent meta-analysis based on 41 cohort studies with 7762 patients, a high YKL-40 serum/plasma level is correlated with a poor prognosis in patients with solid tumors [7]. Previous studies have shown that YKL-40 tissue expression may serve as a useful prognostic biomarker for solid tumors, such as glioma, ovarian cancer, breast cancer, lung cancer, urologic neoplasm, thyroid carcinoma, and gastric cancer [8–14].

Although in CRCs, high serum YKL-40 levels were associated with a poor prognosis [15,16], the prognostic implication of YKL-40 immunohistochemical expression in CRC tissue remains unclear. As YKL-40 is expressed in stromal components as well as tumor cells, it may be insufficient to evaluate T-YKL-40 expression alone in the clinicopathological significances of YKL-40 [4]. High tumor-infiltrating lymphocytes (TILs) correlate with favorable prognosis in CRC, and programmed death ligand-1 (PD-L1)-targeted therapy may enhance sensitivity to conventional cancer therapy [17,18]. Therefore, an investigation of the impact of YKL-40 expression in tumor cells and stromal components is required.

This study aimed to elucidate the clinicopathological and prognostic implications of YKL-40 immunohistochemical expression through human CRC tissue. We evaluated YKL-40 expression in tumor and immune (T- and I-YKL-40) cells by immunohistochemistry. The correlations between YKL-40 expression and PD-L1 and TILs were evaluated. A detailed analysis based on TILs was performed to elucidate the clinicopathological and prognostic implications of YKL-40 expression.

## 2. Materials and Methods

### 2.1. Patients

Between 1 January 2001, and 31 December 2010, 265 patients who had undergone surgical resection for CRC at the Eulji University Medical Center were enrolled in this study. We reviewed their medical charts, pathological records, and glass slides to assess the following clinicopathological characteristics: age; sex; tumor size; tumor location; tumor differentiation; vascular, lymphatic, and perineural invasion; depth of tumor; lymph node metastasis; metastatic lymph node ratio; distant metastasis; and pathologic tumor node metastasis (pTNM) stage. We evaluated these cases according to the 8th Edition of the American Joint Cancer Committee TNM classifications [19]. This protocol was reviewed and approved by the Institutional Review Board of Eulji University Hospital (Approval No. EMCS 2019-07-013). Clinical outcomes were followed from the date of surgery to either the date of death or recurrence, resulting in a follow-up period ranging from 0 to 60 months.

### 2.2. Tissue Array Method and Immunohistochemical Staining

Five array blocks containing a total of 265 resected colorectal cancer tissue cores obtained from the patients were prepared. Briefly, we collected tissue cores (2 mm in diameter) from individual paraffin-embedded CRC tissues (donor blocks) using a trephine and arranged them in recipient paraffin blocks, as previously described [20]. The staining results of the various intratumoral areas in these tissue-array blocks were highly consistent. We chose a core from each case for analysis. An adequate case was defined as a tumor occupying more than 10% of the core area. Each block contained internal controls consisting of non-neoplastic colon tissue.

Sections (4 μm thickness) for immunohistochemical analysis were cut from each tissue-array block, deparaffinized, and dehydrated. For antigen retrieval, we treated the sections with 0.01 M citrate buffer (pH 6.0) for 5 min in a microwave oven, followed by 3% $H_2O_2$ treatment to quench endogenous peroxidase. We treated the sections with normal serum from the host animal that produced the secondary antibody to block non-specific binding. We then incubated the sections with anti-YKL-40 (Cell Signaling Technology, Beverly, MA, USA), anti-PD-L1 (clone SP263; Ventana Medical Systems, Inc., Tucson, AZ, USA), anti-CD3 (Leica Biosystems, Newcastle Upon Tyne, UK), and anti-CD8 (Leica Biosystems) antibodies.

We carried out immunohistochemical staining following a compact polymer method using a VENTANA BenchMark XT autostainer (Ventana Medical Systems, Inc.). We performed visualization by treating the samples using an OptiView universal 3,3′-diaminobenzidine kit (Ventana Medical Systems, Inc.). We used a negative control stain without a primary antibody to confirm the reaction specificity of each antibody. We lightly counterstained all the immunostained sections with Mayer's hematoxylin. Immunohistochemical staining for YKL-40 and PD-L1 were detected in the cytoplasm and cell membrane, respectively.

Positivity for YKL-40 was defined as moderate and strong cytoplasmic expression in any percentage of positively stained cells. The weak positivity was defined as having similar intensity with a non-specific positivity in the background. Positive expression of YKL-40 showed more intensive expression than non-specific positivity in the background. Positivity for PD-L1 was defined as membranous expression of any intensity in ≥10% of tumor or immune cells, regardless of the intensity of PD-L1 expression.

### 2.3. Determination of Immunoscore

To evaluate the impact of TILs, we determined the immunoscores by immunohisto-chemical staining for CD3 and CD8. We scanned all the immunohistochemically stained slides for CD3 and CD8 using a Pannoramic MIDI II slide scanner (3DHISTECH, Budapest, Hungary). We captured images from two regions—the core of the tumor (CT) and invasive margin (IM)—using the CaseViewer 2.0 software (3DHISTECH). CD3- and CD8-immunoreactive lymphocytes were identified from the captured images using NIH Image Analysis software (version 1.6.0; National Institutes of Health, Bethesda, Maryland, USA) after setting a consistent intensity threshold. The CD3- and CD8-immunoreactive lymphocytes were expressed as pixels in each region. Using the cut-off (median value), the pixel value of each case was classified to high (score 1) and low (score 0). Immunoscores were defined as the sum of the scores of two regions and are divided into high (score 3–4) and low (score 0–2) scores [21].

### 2.4. Statistical Analysis

Statistical analyses were performed using SPSS version 22.0 software (IBM Co., Chicago, IL, USA). The significance of the correlation between the YKL-40 expression and the clinicopathological characteristics was determined using the Chi-squared ($\chi^2$) test (two-sided). The comparisons between the YKL-40 expression and age, tumor size, or metastatic lymph node ratio were analyzed using the two-tailed Student's t test. Survival curves were estimated using the Kaplan–Meier product-limit method, and differences between the survival curves were determined to be significant based on the log-rank test. In addition, the prognostic implication of YKL-40 was evaluated using the Cox regression test. The results were considered statistically significant at $p < 0.05$.

## 3. Results

### 3.1. Clinicopathological Significance of YKL-40 Expression in Colorectal Cancers

Representative images of YKL-40 expression in the tumor and immune cells of CRC tissues are shown in Figure 1. I-YKL-40 and T-YKL-40 were expressed in 64.2% and 22.6% of 265 CRC samples, respectively. Well and moderately differentiated tumors significantly correlated with the I-YKL-40 expression ($p = 0.047$), but not T-YKL-40 expression ($p = 0.576$). However, there was no significant correlation between YKL-40 and the other clinicopathological characteristics (Table 1).

Next, we investigated the correlation between YKL-40 and PD-L1 expression. I-YKL-40 expression significantly correlated with PD-L1 expression in immune cells ($p = 0.020$; Table 2). However, there was no significant correlation between the I-YKL-40 expression and PD-L1 expression in tumor cells ($p = 0.673$). T-YKL-40 expression was not significantly correlated with PD-L1 expression in tumor and immune cells ($p = 0.240$ and $p = 0.846$, respectively). In addition, there was no significant correlation between the I- and T-YKL-40 expression and the immunoscore ($p = 0.447$ and $p = 0.852$, respectively).

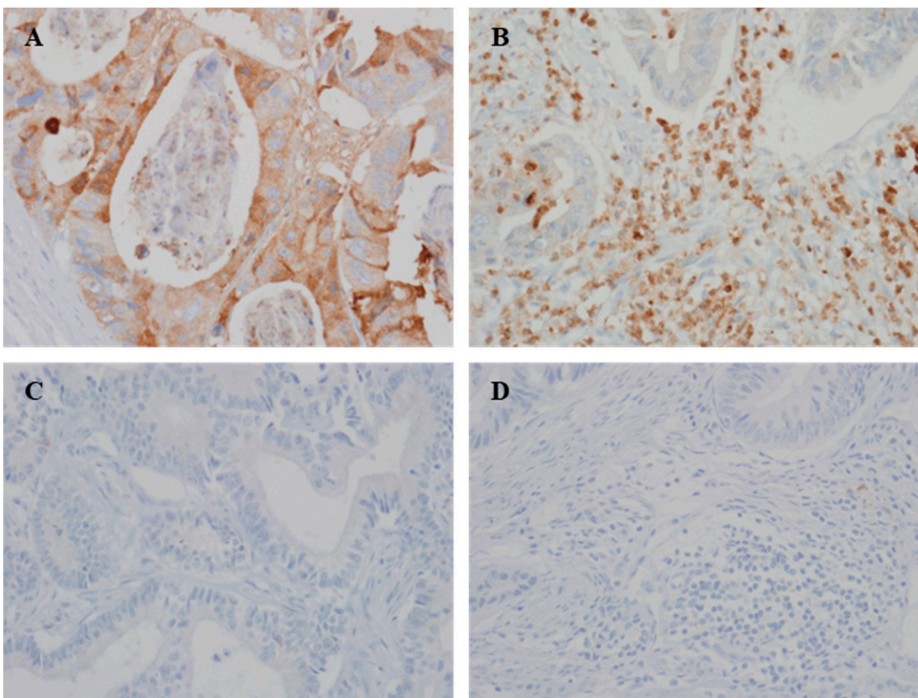

**Figure 1.** Representative immunohistochemical staining images of YKL-40. (**A**,**C**) Positive and negative YKL-40 expression in tumor cells. (**B**,**D**) Positive and negative YKL-40 expression in immune cells (×400).

**Table 1.** Correlation between YKL-40 expression and clinicopathological parameters in colorectal cancers.

| | YKL-40 Expression in Immune Cells | | *p*-Value | YKL-40 Expression in Tumor Cells | | *p*-Value |
|---|---|---|---|---|---|---|
| | **Positive** | **Negative** | | **Positive** | **Negative** | |
| Total (*n* = 265) | 170 (64.2) | 95 (35.8) | | 60 (22.6) | 205 (77.4) | |
| Age (years) | 62.71 ± 13.03 | 65.02 ± 12.63 | 0.162 | 63.85 ± 11.71 | 63.44 ± 13.27 | 0.831 |
| Sex | | | | | | |
| Male | 84 (62.7) | 50 (37.3) | 0.615 | 29 (21.6) | 105 (78.4) | 0.694 |
| Female | 86 (65.6) | 45 (45.8) | | 31 (23.7) | 100 (76.3) | |
| Tumor size | | | | | | |
| ≤5 cm | 68 (64.2) | 38 (35.8) | 1 | 25 (23.6) | 81 (76.4) | 0.764 |
| >5 cm | 102 (64.2) | 57 (35.8) | | 35 (22.0) | 124 (78.0) | |
| Tumor size (cm) | 5.48 ± 2.07 | 5.42 ± 2.10 | 0.84 | 5.52 ± 2.04 | 5.45 ± 2.10 | 0.815 |
| Location of tumor | | | | | | |
| Right colon | 78 (60.9) | 50 (39.1) | 0.292 | 32 (25.0) | 96 (75.0) | 0.375 |
| Left colon and rectum | 92 (67.2) | 45 (32.8) | | 28 (20.4) | 109 (79.6) | |
| Tumor differentiation | | | | | | |
| Well or Moderate | 141 (67.1) | 69 (32.9) | 0.047 | 46 (21.9) | 164 (78.1) | 0.576 |
| Poorly | 29 (52.7) | 26 (47.3) | | 14 (25.5) | 41 (74.5) | |
| Vascular invasion | | | | | | |
| Present | 13 (54.2) | 11 (45.8) | 0.285 | 5 (20.8) | 19 (79.2) | 0.824 |
| Absent | 157 (65.1) | 84 (34.9) | | 55 (22.8) | 186 (77.2) | |
| Lymphatic invasion | | | | | | |
| Present | 39 (55.7) | 31 (44.3) | 0.086 | 16 (22.9) | 54 (77.1) | 0.96 |
| Absent | 131 (67.2) | 64 (32.8) | | 44 (22.6) | 151 (77.4) | |

**Table 1.** *Cont.*

| | YKL-40 Expression in Immune Cells | | *p*-Value | YKL-40 Expression in Tumor Cells | | *p*-Value |
|---|---|---|---|---|---|---|
| | **Positive** | **Negative** | | **Positive** | **Negative** | |
| Perineural invasion | | | | | | |
| Present | 26 (59.1) | 18 (40.9) | 0.443 | 7 (15.9) | 37 (84.1) | 0.243 |
| Absent | 144 (65.2) | 77 (34.8) | | 53 (24.0) | 168 (76.0) | |
| pT stage | | | | | | |
| pT1-2 | 29 (70.7) | 12 (29.3) | 0.339 | 13 (31.7) | 28 (68.3) | 0.131 |
| pT3-4 | 141 (62.9) | 83 (37.1) | | 47 (21.0) | 177 (79.0) | |
| Lymph node metastasis | | | | | | |
| Present | 94 (64.8) | 51 (35.2) | 0.801 | 37 (25.5) | 108 (74.5) | 0.219 |
| Absent | 76 (63.3) | 44 (36.7) | | 23 (19.2) | 97 (80.8) | |
| Distant metastasis | | | | | | |
| Present | 14 (48.3) | 13 (51.7) | 0.059 | 6 (20.7) | 23 (79.3) | 0.79 |
| Absent | 156 (66.1) | 80 (33.9) | | 54 (22.9) | 182 (77.1) | |
| pTNM stage | | | | | | |
| I-II | 74 (64.3) | 41 (35.7) | 0.953 | 23 (24.7) | 113 (75.3) | 0.368 |
| III-IV | 96 (64.0) | 54 (36.0) | | 37 (20.0) | 92 (80.0) | |

Numbers in parentheses represent percentage; pT, pathologic tumor; and pTNM, pathologic tumor node metastasis.

**Table 2.** Correlation between YKL-40 expression and various parameters in colorectal cancers.

| | YKL-40 Expression in Immune Cells | | *p*-Value | YKL-40 Expression in Tumor Cells | | *p*-Value |
|---|---|---|---|---|---|---|
| | **Positive** | **Negative** | | **Positive** | **Negative** | |
| PD-L1 in tumor cells | | | | | | |
| Positive | 17 (68.0) | 8 (32.0) | 0.673 | 8 (31.0) | 17 (68.0) | 0.240 |
| Negative | 153 (63.7) | 87 (36.3) | | 52 (41.9) | 188 (78.3) | |
| PD-L1 in immune cells | | | | | | |
| Positive | 37 (78.7) | 10 (21.3) | 0.020 | 10 (21.3) | 37 (78.7) | 0.846 |
| Negative | 132 (60.8) | 85 (39.2) | | 49 (22.6) | 168 (77.4) | |
| Immunoscore | | | | | | |
| High (score 3–4) | 70 (64.8) | 38 (28.8) | 0.852 | 27 (28.4) | 81 (75.0) | 0.447 |
| Low (score 0–2) | 100 (63.7) | 57 (43.8) | | 33 (19.4) | 124 (79.0) | |

Numbers in parentheses represent the percentage; PD-L1, programmed death-ligand 1.

### 3.2. Correlation between YKL-40 Expression and Prognosis

I-YKL-40 expression significantly correlated with worse overall survival but not recurrence-free survival ($p = 0.047$ and $p = 0.080$, respectively; Figure 2). However, there was no significant correlation between the T-YKL-40 expression and survival. In subgroup analysis based on the pTNM stage, the YKL-40 expression of immune cells had the important prognostic role in stage III (overall survival; $p = 0.048$ and recurrence-free survival; $p = 0.034$), but not other stages. In the multivariate analysis, I-YKL-40 expression had significant prognostic roles (overall survival; hazard ratio 1.683, 95% CI 1.119–2.529, $p = 0.012$ and recurrence-free survival; hazard ratio 1.577, 95% CI 1.074–2.316, $p = 0.020$).

However, there was no significant difference of survival according to T-YKL-40 expression in the multivariate analysis (overall survival; $p = 0.199$ and recurrence-free survival; $p = 0.336$). In addition, in the multivariate analysis, the vascular and perineural invasion, pT stage, and distant metastasis were significantly correlated with worse prognosis.

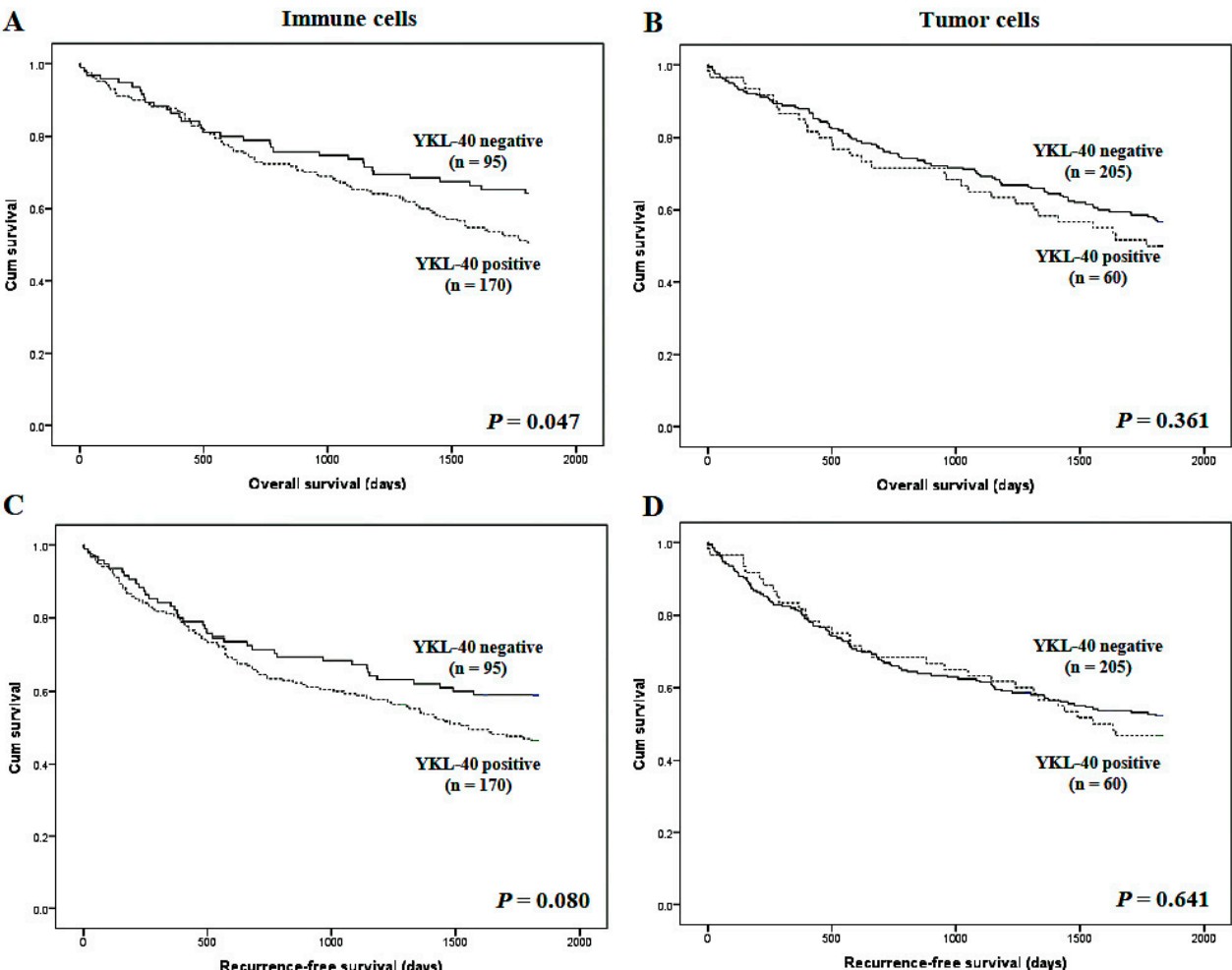

**Figure 2.** Kaplan–Meier analysis of patient survival according to YKL-40 expression in overall cases. (**A**) Overall survival according to YKL-40 expression in immune cells. (**B**) Overall survival according to YKL-40 expression in tumor cells. (**C**) Recurrence-free survival according to YKL-40 expression in immune cells. (**D**) Recurrence-free survival according to YKL-40 expression in tumor cells.

We also analyzed the prognostic roles of YKL-40 expression in high and low immunoscore subgroups. Interestingly, patients with I-YKL-40 expression had worse overall and recurrence-free survival than patients without I-YKL-40 expression in the subgroup with high immunoscores ($p = 0.008$ and $p = 0.042$, respectively; Figure 3). In contrast, there was no significant difference in the survival between patients with and without T-YKL-40 expression in CRCs with high immunoscores. In low immunoscore subgroups, there was no significant correlation between the I- and T-YKL-40 expression and survival.

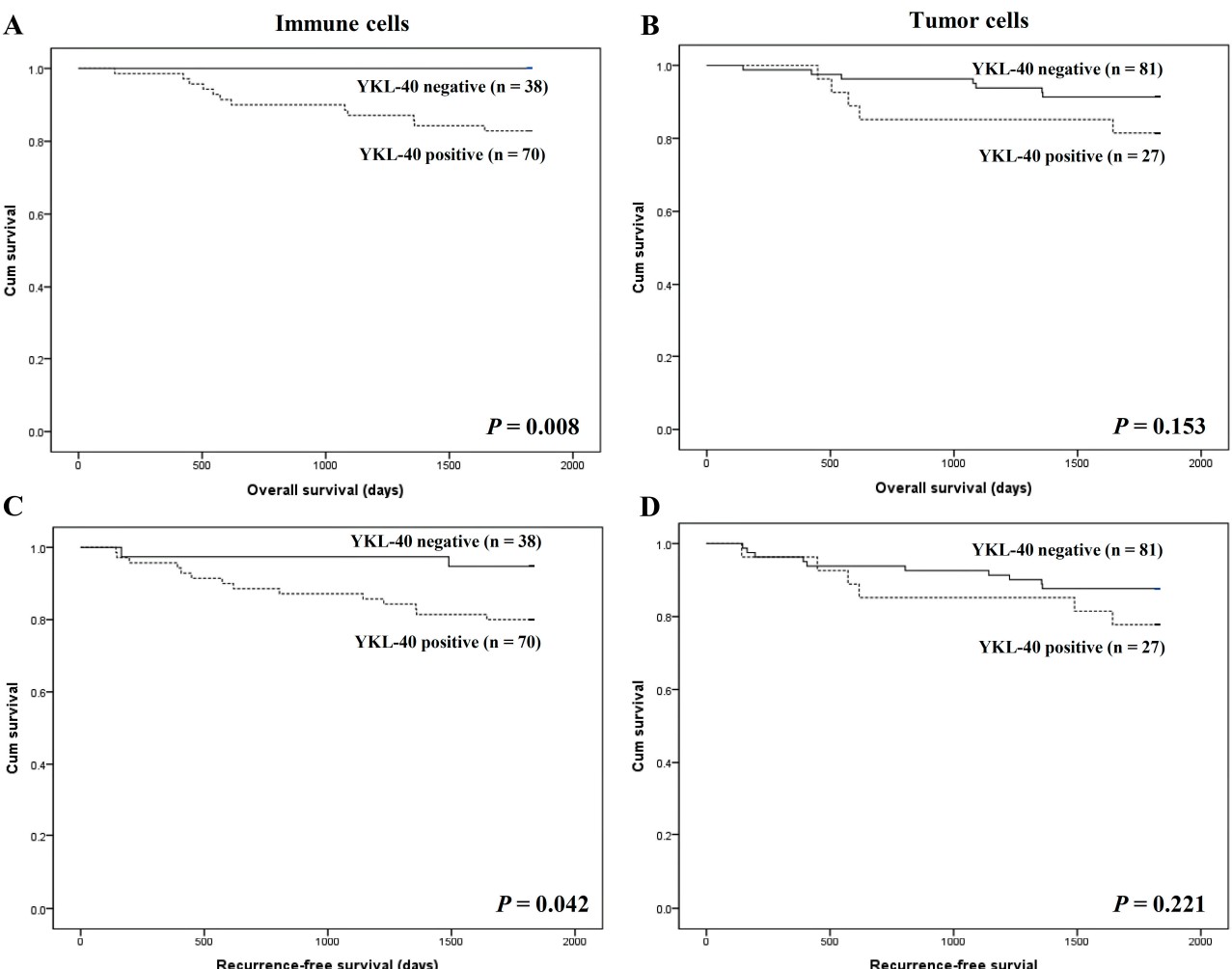

**Figure 3.** Kaplan–Meier analysis of patient survival according to YKL-40 expression in colorectal cancer cells of patients with high immunoscores. (**A**) Overall survival according to YKL-40 expression in immune cells. (**B**) Overall survival according to YKL-40 expression in tumor cells. (**C**) Recurrence-free survival according to YKL-40 expression in immune cells. (**D**) Recurrence-free survival according to YKL-40 expression in tumor cells.

## 4. Discussion

In the previous studies, the prognostic impacts of YKL-40 were mainly evaluated through the serum/plasma. However, the prognostic implication of YKL-40 expression in CRC tissue is not fully understood. To the best of our knowledge, the present study is the first attempt to evaluate the clinicopathological significance of YKL-40 expression according to cell types in human CRC tissues. In addition, the prognostic impacts of YKL-40 expression according to the immunoscore were evaluated.

Compared to normal cells, YKL-40 expression is high in various malignant tumors, including colorectal, gastric, hepatocellular, pancreatic, breast, non-small cell lung, and ovarian cancers [22]. Liu et al. compared the expression of YKL-40 in tumor and adjacent normal tissues [23]. The YKL-40 expression rates were 93.33% and 46.66% in 15 colon cancer and 15 adjacent normal tissues, respectively [23]. In our study, the T-YKL-40 expression rate was 22.6%. This discrepancy of YKL-40 expression rate may be caused by different criteria of interpretation. In Liu's report, cases with weak positivity were classified as positive cases.

In the present study, cases with weak positivity were considered to be negative, unlike in the previous study. If cases with weak positivity are considered negative, the positive expression rate is re-estimated as 60.0% and 13.3% in tumor and adjacent normal tissues,

respectively [23]. Previous studies have reported that YKL-40 is expressed in the tissue of approximately 20–40% of patients with various solid tumors, which is consistent with our findings [10,14,24,25]. In the present study, we investigated YKL-40 expression in both immune and tumor cells.

To the best of our knowledge, no previous studies analyzed the expression of I-YKL-40 in CRC patients. The positive expression rate of YKL-40 in immune cells was 64.2% in CRC patients. In addition, we first investigated the correlation between I-YKL-40 expression and clinicopathological characteristics. Although the I-YKL-40 expression significantly correlated with well or moderately differentiated tumors, there was no significant correlation with other clinicopathological characteristics.

A previous meta-analysis showed that elevated circulating levels of YKL-40 were significantly related to poor survival in cancer patients [7]. Certain studies have examined the prognostic implication of YKL-40 tissue expression in patients with solid tumors, but these studies have yielded inconsistent results [8–14,26–28]. In the case of CRC, several studies have shown that elevated serum YKL-40 concentration is associated with poor prognosis [15,16,29]. Tumor cell-derived YKL-40 is closely connected to VEGF-independent tumor angiogenesis, which is a critical factor in tumor progression [25,30]. A study revealed that a blockade of YKL-40 function impeded tumor angiogenesis and progression both in vivo and in vitro [31].

They suggested that monoclonal antibodies against YKL-40 inhibited the activation of VEGF receptor 2 and intracellular signaling MAP kinase [31]. In our cases, there was a significant correlation between the I-YKL-40 expression and high microvessel density ($p$ = 0.011; data not shown). Interestingly, it is not fully documented that the tissue expression of YKL-40 is inversely associated with clinical outcomes in patients with CRC. In a previous study, Kaplan–Meier analysis showed that the level of YKL-40 in CRC tissue correlated with overall survival [23].

However, because the positive expression rate of YKL-40 was 89.7% in tumor tissue, its prognostic role can be limited. In addition, previous studies have not addressed other tumor components, such as TILs. Our study suggests that a useful prognostic marker in patients with CRC may be I-YKL-40 expression rather than T-YKL-40 expression. In the present study, I-YKL-40 expression was significantly associated with poor overall survival in enrolled patients.

The immunoscore is introduced as a prognostic tool based on CD3- and CD8-immunoreactive lymphocytes at the core of the tumor and invasive margin [32,33]. In our previous study, a high immunoscore was significantly associated with better survival in patients with CRC [34]. The role of YKL-40 in TILs and their clinical implications has not yet been fully elucidated. In vivo, YKL-40 is expressed in activated T cells and regulates IFNγ sensitivity in Th1 and cytotoxic T-lymphocytes [35]. YKL-40 can attenuate T-cell activation and tumor growth. However, YKL-40 was associated with the suppression of TILs. Therefore, the correlation between YKL-40 expression and TILs may be important in the interpretation of the clinicopathological significance of YKL-40.

We investigated the correlation between I-YKL-40 and prognosis according to the immunoscore. Interestingly, our study demonstrated that I-YKL-40 might be an important factor for stratifying CRC patients with high immunoscores. Neither T-YKL -40 nor I-YKL-40 expression correlated with the immunoscore, whereas higher I-YKL-40 expression, but not T-YKL-40, significantly correlated with a poor prognosis in CRC patients with high immunoscores. Our results suggest that prognostic stratification with YKL-40 and TILs may be applied to minimizing the effects of tumor heterogeneity in CRCs.

YKL-40 has been a promising therapeutic target in cancers [31,36]. An in vitro study found that theophylline had a suppressive effect on human rectal cancer cells by inhibiting YKL-40 expression [24]. An in vivo study revealed that a neutralizing antibody against YKL-40 inhibited tumor growth, angiogenesis, and progression in xenograft models [31]. A recent study showed that a high expression of YKL-40 could sensitize tumor cells to an anti-angiogenic drug (cetuximab) in patients with CRC [23]. In addition, our study

demonstrated a significant correlation between I-YKL-40 and PD-L1 expression, which may provide novel therapeutic strategies for CRC treatment.

A recent meta-analysis revealed that high expression of PD-L1 was significantly correlated with a poor prognosis of patients with CRC [37]. PD-L1 targeted therapy had a major advantage of increasing the sensitivity to conventional cancer therapies [17]. Previous studies suggested that the simultaneous administration of anti-YKL-40 and anti-PD-1 antibodies showed antitumor effects in vivo and in vitro [38–40]. Although the direct correlation between YKL-40 and PD-L1 expressions is unclear, we expect that a combination of anti-YKL-40 and anti PD-L1 antibodies may enhance the potential of immunotherapies and sensitivity to anti-angiogenic therapies in CRCs.

## 5. Conclusions

In conclusion, I-YKL-40 expression is significantly correlated with tumor differentiation and I-PD-L1 in CRCs. In addition, there was a significant correlation between poor survival outcome and I-YKL-40 expression, but not T-YKL-40 expression, in CRC patients with a high immunoscores. I-YKL-40, but not T-YKL-40, could be considered a useful predictor of poor prognosis in CRC for the stratification of patients with high immunoscores.

**Author Contributions:** Conceptualization, B.K.S. and I.H.O.; methodology, J.-S.P.; software, J.-S.P.; validation, B.K.S., J.-S.P., and I.H.O.; formal analysis, J.-S.P.; investigation, I.H.O.; resources, B.K.S.; data curation, J.-S.P.; writing—original draft preparation, I.H.O.; writing—review and editing, J.-S.P. and B.K.S.; supervision, B.K.S.; project administration, B.K.S., J.-S.P., and I.H.O.; funding acquisition, B.K.S. All authors have read and agreed to the published version of the manuscript.

**Funding:** This research was funded by Eulji University, grant number 2019EMBRISN0004.

**Institutional Review Board Statement:** This study was conducted according to the guidelines of the Declaration of Helsinki, and approved by the Institutional Review Board of the Eulji University Hospital (Approval No. EMC 2020-09-011).

**Informed Consent Statement:** Not applicable.

**Data Availability Statement:** No new data were created or analyzed in this study. Data sharing is not applicable to this article.

**Conflicts of Interest:** The authors declare no conflict of interest.

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
