# Peer review of "Prognostic Impact of YKL-40 Immunohistochemical Expression in Patients with Colorectal Cancer"

_curroncol, doi:10.3390/curroncol28040274_

Round 1

Reviewer 1 Report

YKL-40 or Chitinase-3-like protein 1 is a secreted glycoprotein encoded by the CHI3L1 gene, its biological function is not clear.  However, it is expressed by a variety of cells such as macrophages, vascular smooth muscle cells, and tumor cells.  Its expression is associated with pathogenic processes related to inflammation, extracellular tissue remodeling, fibrosis and solid carcinomas.  The authors in this research paper, studied the YKL-40 expression by performing immunohistochemistry and investigated the clinicopathological and prognostic impact of YKL-40 expression in tumor cells (T-YKL-40) and infiltrated immune cells in (I-YKL-40) in colorectal cancer (CRC).  They also evaluated the correlation between YKL-40 and PD-L1 expression. Their study suggested that the PD-L1 expression in the immune cells significantly correlated I-YKL-40expression but not T-YKL-40 expression.  I-YKL-40 expression was significantly correlated with a worse overall survival rate but not recurrence free survival.  There was no significant correlation between T-YKL-40 expression and survivals.  In CRCs with high immunoscore, patients with I-YKL-40 expression demonstrated worse overall and recurrence-free survival than those without I-YKL-40 expression.  They suggest that I-YKL-40 expression is significantly correlated with tumor differentiation and PD-L1 expression in immune cells.   

Author Response

YKL-40 or Chitinase-3-like protein 1 is a secreted glycoprotein encoded by the CHI3L1 gene, its biological function is not clear. However, it is expressed by a variety of cells such as macrophages, vascular smooth muscle cells, and tumor cells. Its expression is associated with pathogenic processes related to inflammation, extracellular tissue remodeling, fibrosis and solid carcinomas. The authors in this research paper, studied the YKL-40 expression by performing immunohistochemistry and investigated the clinicopathological and prognostic impact of YKL-40 expression in tumor cells (T-YKL-40) and infiltrated immune cells in (I-YKL-40) in colorectal cancer (CRC). They also evaluated the correlation between YKL-40 and PD-L1 expression. Their study suggested that the PD-L1 expression in the immune cells significantly correlated I-YKL-40expression but not T-YKL-40 expression. I-YKL-40 expression was significantly correlated with a worse overall survival rate but not recurrence free survival. There was no significant correlation between T-YKL-40 expression and survivals. In CRCs with high immunoscore, patients with I-YKL-40 expression demonstrated worse overall and recurrence-free survival than those without I-YKL-40 expression. They suggest that I-YKL-40 expression is significantly correlated with tumor differentiation and PD-L1 expression in immune cells.

Response:

                  Thank you for the careful review.

Reviewer 2 Report

The revision did not enhance the value of the article. The multivariable analysis did not add anything meaningful. The last 2 lines of the multivariable analysis table are confusing- could not tell what that meant.

Author Response

The revision did not enhance the value of the article. The multivariable analysis did not add anything meaningful. The last 2 lines of the multivariable analysis table are confusing- could not tell what that meant.

Response:

                  As reviewer’s recommendation, we performed the additional multivariate analysis.

Overall survival

Recurrence-free survival

Hazard ratio

P-value

Hazard ratio

P-value

Sex, male vs. female

0.799 (0.552, 1.156)

0.233

0.743 (0.522, 1,057)

0.099

Size group, >5 vs. ≤5cm

1.456 (0.960, 2.207)

0.077

1.471 (0.987, 2.192)

0.058

Site, right vs. left

1.062 (0.726, 1.553)

0.757

1.275 (0.888, 1.832)

0.188

Vascular invasion, present vs. absent

1.622 (0.841, 3.129)

0.149

1.933 (1.036, 3.606)

0.038

Lymphatic invasion, present vs. absent

1.191 (0.755, 1.877)

0.453

1.011 (0.658, 1.554)

0.960

Perineural invasion, present vs. absent

1.880 (1.168, 3.026)

0.009

1.930 (1.237, 3.012)

0.004

pT stage, high vs. low

2.511 (1.117, 5.645)

0.026

2.921 (1.311, 6.509)

0.009

Lymph node metastasis, present vs. absent

1.212 (0.793, 1.852)

0.374

1.394 (0.932, 2.084)

0.105

Distant metastasis, present vs. absent

3.494 (2.129, 5.735)

< 0.001

2.804 (1.720, 4.573)

< 0.001

According to the results, I-YKL-40 expression, but not T-YKL-40 expression, was significantly correlated with worse overall and recurrence-free survival.

We added the results of multivariate analysis for other prognostic factors in the revised manuscript as below:

In the multivariate analysis, I-YKL-40 expression had significant prognostic roles (overll survival; hazard ratio 1.683, 95% CI 1.119-2.529, P = 0.012 and recurrence-free survival; hazard ratio 1.577, 95% CI 1.074-2.316, P = 0.020). However, there was no significant difference of survival according to T-YKL-40 expression in the multivariate analysis (overall survival; P = 0.199 and recurrence-free survival; P = 0.336). In addition, in the multivariate analysis, the vascular and perineural invasion, pT stage, and distant metastasis were significantly correlated with worse prognosis.

This manuscript is a resubmission of an earlier submission. The following is a list of the peer review reports and author responses from that submission.

Round 1

Reviewer 1 Report

YKL-40 or Chitinase-3-like protein 1 is a secreted glycoprotein encoded by the CHI3L1 gene, its biological function is not clear.  However, it is expressed by a variety of cells such as macrophages, , vascular smooth muscle cells, and tumor cells.  Its expression is associated with pathogenic processes related to inflammation, extracellular tissue remodeling, fibrosis and solid carcinomas.  The authors in this research paper, studied the YKL-40 expression by performing immunohistochemistry and investigated the clinicopathological and prognostic impact of YKL-40 expression in tumor cells (T-YKL-40) and infiltrated immune cells in (I-YKL-40) in colorectal cancer (CRC).  They also evaluated the correlation between YKL-40 and PD-L1 expression. Their study suggested that the PD-L1 expression in the immune cells significantly correlated I-YKL-40 expression but not T-YKL-40 expression.  I-YKL-40 expression was significantly correlated with a worse overall survival rate but not recurrence free survival.  There was no significant correlation between T-YKL-40 expression and survivals.  In CRCs with high immunoscore, patients with I-YKL-40 expression demonstrated worse overall and recurrence-free survival than those without I-YKL-40 expression.  They suggest that I-YKL-40 expression is significantly correlated with tumor differentiation and PD-L1 expression in immune cells.   

Minor comments

  1.  Please expand in the discussion on the relationship between YKL-40 expression and PD-L1 expression.  Does YKL-40 regulates PD-L1? Provide details.
  2. How is YKL-40 expression connected to VEGF?  Provide an explanation in the discussion.

Reviewer 2 Report

The authors retrospectively analyzed CRC tumor tissues to evaluate if a correlation exists between YKL-40 expression in the patients' tumor/immune cells and prognosis. This manuscript has many flaws, and the major flaws are in the methodology section as outlined below-

  1. Survival curves have been constructed without any regard to the TNM staging system. As TNM staging is one of the most powerful prognostic factors for patients with CRC, a biomarker analysis in a group of patients with heterogeneous TNM stages is not scientifically valid (unless supported by a robust multivariable analysis). The authors should perform this analysis in a homogeneous group of patients, for example, patients with low-risk stage III or average risk stage II colon cancer.
  2. Furthermore, the authors did not perform a multivariable analysis with the known prognostic factors.
  3. No information provided on the treatment given to the patients.

Reviewer 3 Report

Paper reviewed the YKL-40 expression in colorectal cancer specimens and their prognostic implication, correlation to the immune scores, PD-L1 expression and prognosis were evaluated. Immune YKL-40 expression correlated to the well and moderately differentiated tumors as per clinicopathological significance. With high immunoscores Immune YKL-40 was associated with poor survival. No clear practice changing implications at this time but can be a future target in the treatment of colorectal cancer.